# Exosomes-Based Nanomedicine for Neurodegenerative Diseases: Current Insights and Future Challenges

**DOI:** 10.3390/pharmaceutics15010298

**Published:** 2023-01-16

**Authors:** Amanda Cano, Álvaro Muñoz-Morales, Elena Sánchez-López, Miren Ettcheto, Eliana B. Souto, Antonio Camins, Mercè Boada, Agustín Ruíz

**Affiliations:** 1Ace Alzheimer Center Barcelona—International University of Catalunya (UIC), 08028 Barcelona, Spain; 2Biomedical Research Networking Centre in Neurodegenerative Diseases (CIBERNED), 28029 Madrid, Spain; 3Institute of Nanoscience and Nanotechnology (IN2UB), 08028 Barcelona, Spain; 4Department of Pharmacy, Pharmaceutical Technology and Physical Chemistry, Faculty of Pharmacy and Food Sciences, University of Barcelona, 08028 Barcelona, Spain; 5Unit of Synthesis and Biomedical Applications of Peptides, IQAC-CSIC, 08034 Barcelona, Spain; 6Department of Pharmacology, Toxicology and Therapeutic Chemistry, Faculty of Pharmacy and Food Sciences, University of Barcelona, 08028 Barcelona, Spain; 7Institute of Neurosciences, University of Barcelona, 08028 Barcelona, Spain; 8Department of Pharmaceutical Technology, Faculty of Pharmacy, University of Porto, 4050-313 Porto, Portugal; 9REQUIMTE/UCIBIO, Faculty of Pharmacy, University of Porto, 4050-313 Porto, Portugal

**Keywords:** exosomes, extracellular vesicles, nanomedicine, nanotechnology, neurodegenerative diseases

## Abstract

Neurodegenerative diseases constitute a group of pathologies whose etiology remains unknown in many cases, and there are no treatments that stop the progression of such diseases. Moreover, the existence of the blood–brain barrier is an impediment to the penetration of exogenous molecules, including those found in many drugs. Exosomes are extracellular vesicles secreted by a wide variety of cells, and their primary functions include intercellular communication, immune responses, human reproduction, and synaptic plasticity. Due to their natural origin and molecular similarities with most cell types, exosomes have emerged as promising therapeutic tools for numerous diseases. Specifically, neurodegenerative diseases have shown to be a potential target for this nanomedicine strategy due to the difficult access to the brain and the strategy’s pathophysiological complexity. In this regard, this review explores the most important biological-origin drug delivery systems, innovative isolation methods of exosomes, their physicochemical characterization, drug loading, cutting-edge functionalization strategies to target them within the brain, the latest research studies in neurodegenerative diseases, and the future challenges of exosomes as nanomedicine-based therapeutic tools.

## 1. Introduction

In recent decades, the improvement in the quality of life in developed countries has led to an increase in life expectancy and, thus, the aged population, which is closely related to the increase in the prevalence of neurodegenerative diseases [1]. Neurodegeneration is defined as progressive damage in neural tissues drifting to an irrecoverable neuronal loss, which impacts cognitive function, motor activity, and the mental impairment of these patients [2]. These pathologies encompass a large variety of disorders with different pathological patterns and clinical manifestations, such as Alzheimer’s disease (AD), epilepsy, Parkinson’s disease (PD), and multiple sclerosis (MS) [3].

One of the main problems involving the success of pharmacological therapies in treating these diseases is the limited passage of exogenous molecules into the central nervous system (CNS) [4]. Restricted drug delivery into the brain is mostly attributed to the blood–brain barrier (BBB). The endothelial cell barrier comprises the main physical barrier where different molecules involved in the transport of substances are found. This penetration restriction stems mainly from the tight junctions between the endothelial cells, the lack of transcellular pathways, especially for hydrophilic drugs, and the unavailability of transport vesicles [5]. Because of these, crossing the BBB has gained much attention within the scientific and health communities. Drugs administered systemically must pass from the blood circulation into the CNS by crossing this physiological barrier, whose fundamental function is to protect the CNS from harmful substances [5].

Exosomes are extracellular vesicles (EVs) loaded with a wide variety of molecules, such as RNA, amino acids, lipids, metabolites, or proteins from their cells of origin [6]. Exosome biogenesis is initiated by plasma membrane invagination of the origin cell and is located in multivesicular bodies (MVBs), which are formed intracellularly [7]. Then, MVBs are transported to the luminal side of the plasma membrane through the microtubule network and the cytoskeleton [7], and exosomes are released to the cytosol by an exocytosis process [7]. Exosomes are involved in a wide variety of biological functions. It is well known that exosomes play a key role in cellular intercommunication, but they also participate in immune responses, the maintenance of synaptic plasticity and neurotransmission, putative functions in human reproduction, pregnancy, and embryonic development, and wound healing, among others [7]. Furthermore, recent evidence has shown that exosomes can easily cross the BBB bidirectionally, thus connecting the central and peripheral compartments [8].

Since exosomes can be transported through the whole body and released into different bodily fluids, they are receiving significant interest as a promising source of biomarkers for many diseases [9]. Thus, the existence of exosomes with specific proteins and surface markers has been identified in different human diseases, including cancers [10], AD [11], sarcoidosis [12], cardiovascular diseases [13], or prion diseases [14], among others. Because of this, circulating exosomes have been proposed as a tool for diagnosing and monitoring brain diseases. Likewise, exosomes have also been proposed as nanomedicine-based strategies for the treatment of neurological diseases, acting as natural drug delivery systems.

Controlled drug delivery systems are designed to overcome the biopharmaceutical limitations of common formulations, such as tablets, capsules, or syrups. These carriers range between 1–1000 nm and can be formulated with synthetic or natural compounds [15]. Likewise, these vehicles possess some specific properties that lead to the release of the drug in a sustained manner, remain in the body for a specified period of time, can be administered locally or systemically, and target the carried drug to the specific site of action [15]. Different nanovehicles commonly used in nanomedicine applications possess these properties, such as liposomes, micelles, nanoparticles, carbon nanotubes, dendrimers, quantum dots, or exosomes [16].

In this review, we will explore the use of exosomes as a nanomedicine-based strategy for neurodegenerative diseases. We will describe the most innovative isolation and fabrication methods, their physicochemical characterization and drug-loading processes, the cutting-edge functionalization strategies to target them within the brain, the latest research on exosomes as therapeutic tools for brain disorders, and the current limitations and future perspectives of exosomes as a nanomedicine-based therapeutic strategy.

## 2. Delivery Systems from Biological Origin

Drug delivery systems aim to improve the pharmacological properties of drugs and achieve maximal therapeutic efficacy and minimal side effects by directing therapeutic cargo to target cells and tissues. However, human bodies are equipped with innate immune defenses, such as the reticuloendothelial system (RES), which rapidly recognizes and destroys foreign objects [17]. Because of that, biological drug carriers are arousing much interest since they can be used to bypass immune surveillance. Cells and their components have the natural ability to sense, integrate, and respond to dynamic environments in vivo, making them an attractive vehicle for the delivery of different therapeutic compounds [18]. Apart from exosomes, biological carriers such as platelets, red blood cells (RBC), or albumin provide several advantages, such as good bioavailability and biodegradability, long circulation time, flexible morphology, and abundant surface ligands [18,19]. Therefore, these carriers represent a feasible solution to overcome the limitations of synthetic nanomaterial-based drug carriers. The most important biological drug carriers are described in Table 1.

## 3. Isolation Methods

Since they can originate from different cell types of different tissues and organs, exosomes possess a high complexity and heterogeneity, which make the isolation process difficult. Several reproducible isolation techniques have been developed depending on exosomes’ different biochemical characteristics, including mass density, size, charge, shape, and surface antigens [20]. However, most current isolation technologies cannot completely separate exosomes in a pure batch, and all of them present some advantages and disadvantages (Table 2). Such results make it difficult to rank the performance of the different methods [21].

### 3.1. Ultracentrifugation

Ultracentrifugation (UC) is one the most commonly used techniques for exosome isolation, being described as the gold standard for exosome separation. Exosome fractions can be obtained depending on their size and density, so it is suitable for the separation of large-volume samples with significant differences in the sedimentation coefficient [22]. When separated by size, the process is divided into two steps: a series of low-medium speed centrifugation to remove the dead cells and large-size extracellular vesicles, and then higher speed centrifugation at 100,000× *g* to obtain the exosomes [22]. When separated by density, the main purpose is to purify the exosomes by using sucrose or iodixanol as a medium. While sucrose cannot effectively separate exosomes and retroviruses, iodixanol enables this separation and the harvest of high-purity exosomes [21]. Although UC is a very common method of EV isolation, it also shows serious limitations if the correct conditions are not exactly met. Particularly, the co-precipitation of aggregated proteins and alterations of EVs functionality can be observed [23,24,25].

### 3.2. Ultrafiltration

The ultrafiltration (UF) process is quite similar to UC. This method usually uses UF membranes with different molecular weight cutoffs to selectively separate samples [26]. Apoptotic bodies and larger microvesicles are removed from the sample’s matrix by applying pressure with a 0.22-μm filter. Importantly, compared to UC, UF can concentrate exosomes by up to 240-fold and possesses a higher sample throughput. In addition, this technique also leads to the high purity of the exosome fractions, adaptability for high-throughput proteomics analysis, and does not have sample volume limitations. However, since many exosomes bind to the filters, the exosomal protein yield is relatively low [27,28].

### 3.3. Size Exclusion Chromatography

Together with UF, size-exclusion chromatography (SEC) separates the exosomes based on the size difference between them and other components in the biological samples. Macromolecules cannot penetrate the gel pores and are eluted with the mobile phase along the gaps, while the small molecules remain in the gel pores and are finally eluted by the same mobile phase. SEC is widely popular in exosome enrichment chiefly because of this elution [23,24,29]. The choice of the exclusion matrix determines the exosome size cutoff. SEC can remove almost 99% of the proteins present in the biological fluid, thus resulting in high-purity fractions of the exosomes without aggregation events [30] while maintaining the biological activity and morphological integrity of the exosomes [24]. SEC can recover around 40–90% of exosomes with high reproducibility but with low protein yields. In addition, several particles above the size cut off, such as low-density LDL or viruses, can be co-isolated [24,29,30]. However, SEC possesses a high throughput (1.5 h in total) [29], which makes it a suitable option for large-scale studies. Furthermore, it has been widely used together with LC-MS for high-throughput exosomal biomarkers analysis [31].

### 3.4. Immunoaffinity Chromatography

Immunoaffinity chromatography (IC) is a separation and purification technology based on the binding affinity of immobilized antibodies to specific ligands on the surface of the exosomes. The binding efficiency is closely related to the matrix carriers, the biological affinity pairs, and the elution conditions [21]. The surface molecules detected by this method should be in high-abundance proportions in the exosome membranes to lead the binding. Monoclonal antibodies are commonly used and immobilized on a solid phase (e.g., magnetic beads). However, other immunocapture assays also use chemical affinity or annexin A5, which binds to phosphatidyl serine residues on the exosomes’ surface [32,33]. This technique presents many advantages. Compared to UC, IC requires a significantly lower sample volume with comparable results. It has high purity and yield, high sensitivity, and strong specificity. In addition, it also can be used for qualitative and quantitative studies. However, IC also presents various disadvantages. The storage conditions of the obtained exosomes are complicated and not suitable for large-scale separation. Likewise, another major limitation of this technique is the batch effect. The non-specific interference adsorption of the matrix will produce interfering proteins, which can lead to low reproducibility for proteomic analysis. Moreover, IC may take several hours for one single enrichment, even with the latest microchannel devices [21].

### 3.5. Polymer Precipitation

Polymer precipitation (PP) was originally used to isolate viruses. It usually uses water-excluding polymers, such as polyethylene glycol (PEG), as a medium to reduce the solubility of exosomes, thus leading to their harvest under a low-speed centrifugation process [34]. Interestingly, PP is more cost-effective than commercial kits and surpasses UC in terms of purity and recovery. PP is relatively easy to carry out, as it is minimally time-consuming and suitable for processing large doses of samples. However, the purity and recovery rate are relatively low, and false positives may be generated [21]. In addition, the polymer produced is difficult to remove, which could impede subsequent functional experimental analysis. Likewise, another limitation is the co-precipitation of abundant non-exosome molecules, such as proteins, and hence this process is not suitable for MS-based proteomics studies [21].

### 3.6. Commercial Kits

There are several commercial kits on the market based on a combination of the above-described isolation technology, mainly immunoaffinity and precipitation. The most-used kits, those with the best purity and isolation rates and easiest handling, are ExoQuick^®^ (System Biosciences, The Bay Area, CA, USA) [35], the exoEasy^®^ Maxi kit (QIAGEN^©^) [36], the Minute™ Hi-Efficiency Exosome Precipitation Reagent (Invent Biotechnology, Eden Prairie, MN, USA) [37], and the MagCapture™ Exosome Isolation Kit PS (FUJIFILM Wako, Tokyo, Japan) [38]. The main advantages of these commercial kits are their easy handling, resulting time saved, high yield, good integrity, and low volume requirements, all of which combine to confer them with the ideal characteristics for implementation in clinical studies. However, due to the uneven extraction effect of the current commercial kits, there is still no kit that can isolate exosomes from a mixture of samples [21]. Likewise, these kits are expensive, and the purity of the exosomes is not always high. Nevertheless, innovation and continuous improvements in their development and performance are currently leading to better performance and outcomes, thus resulting in potential application value.

Although a wide variety of methods for the isolation and purification of exosomes have been developed, all of them present some shortcomings that mean they cannot meet all needs. Consequently, a combination of different isolation methods would likely be the better solution to reduce the sample volume and improve the purity and isolation rates.

## 4. Physicochemical Characterization of Exosomes

Characterization is an important step to validate the isolation process of exosomes. Size, morphology, concentration, homogeneity of population, and surface proteins can differ between different types of EVs (Figure 1) [39,40].

Nanoparticle tracking analysis (NTA) is the gold-standard method for characterizing the size, particle distribution, and concentration of exosomes [41]. This is an optical method that uses a microscope to detect the Brownian motion of individual particles dispersed in a fluid medium by using image analysis [42]. This biophysical approach can also employ fluorescence labeling to detect the antigens present on the exosomes [43]. Importantly, the samples can be completely recovered after measuring.

Dynamic light scattering (DLS) is a biophysical optical approach commonly used to analyze the average size and distribution in a suspension of particles [44]. It is based on the detection of the light-scattering changes produced by particles when they cross through light due to their Brownian movements. However, this technique is not useful when larger EVs are present in the exosome solutions [45].

Transmission electron microscopy (TEM) and atomic force microscopy (AFM) are used to determine exosomes’ morphology and size [46]. TEM is a widely used electron microscopy approach suited for high magnification and resolution imaging [47]. In the case of exosomes, cryo-TEM is commonly used since it is a more accurate technique due to its use of liquid nitrogen, which prevents the effects of dehydration and fixation and provides images without isolation artifacts [48]. Concurrently, AFM performs surface scanning with the tip of a cantilever beam to achieve sub-nanometer resolution imaging [49]. The main advantages of AFM include its easy handling and its ability to measure samples in their natural state [50].

The analysis of exosomes’ distinctive surface proteins is usually performed by using Western blot and flow cytometry. Western blotting provides a qualitative result relating to the presence or absence of marker proteins. The Bradford assay and Pierce BCA can be used to perform the quantitative assay of the total protein content in the exosome samples [51]. Flow cytometry is then used to evaluate and analyze the cellular origin of single exosomes [52]. Due to their small size, the exosomes are linked to beads and then subjected to flow cytometry using fluorescence-activated cell sorting for analysis [39]. ELISAs have also been explored for the detection and quantification of exosomal protein markers. However, ELISA plate-based techniques require a large number of samples and do not have high sensitivity [46,53].

Finally, flow cytometry is a laser-based technique used to detect bead-bound exosomes, which leads to the characterization of different exosome populations by using specific antibodies to target EVs. This technique does not require the isolation or concentration of the exosomes prior to capturing, uses small volumes of samples, and reduces the overall sample processing time [54,55].

## 5. Exosomes Surface Functionalization for Brain Delivery

Exosomes have the innate ability to cross the BBB [56,57]. However, it has been demonstrated that when exosomes are administered externally (i.e., intravenously), a large fraction of systemically injected exosomes quickly become trapped in hepatic or splenic tissues due to their specialized subsets of phagocytic immune cells and extensive capillary network [58]. Because of this, many techniques have been developed to target these vesicles to the brain and improve their penetration through the BBB [59,60,61] (Figure 2). The conjugation of specific ligands to the surface of exosomes increases the interaction with the target cells, and the addition of labeled fluorescent dyes or radioactive MRI agents is an efficient method for in vivo tracking [62]. This conjugation process can be achieved through different techniques.

### 5.1. Click Chemistry

Copper-catalyzed azide cycloaddition (click chemistry) is one of the most commonly used techniques for exosome surface modification [63]. Compared with traditional chemical reactions, click chemistry has many advantages: a fast reaction time, high specificity, compatibility in aqueous buffer [64], and the conjugation reaction does not affect the size of the exosomes, their uptake, or the absorption of cells [62]. Likewise, click chemistry is very suitable for conjugation between chemical molecules and biomolecules on the surface of exosomes. Through this method, fluorescence markers, radioactivity tracers, or MRI contrast agents can be added to the surface to be tracked in vivo and analyze the biodistribution of exosomes [59]. This method is based on serial chemical reactions to attach all of the substrates to the biomolecules [62,65]. Some popular click chemistry approaches are the bifunctional PEG linker [66], biorthogonal copper-free click chemistry [62], the avidin–biotin complex [67], and the EDC/NHS reaction, which leads to the direct attachment of the ligand to the exosome’s surface [68].

### 5.2. Endogen Receptors

The application of a natural receptor on exosomes’ luminal surface is widely used to increase the exosome transcytosis rate through the BBB. In this sense, TfR, INSR, and especially LDLRs are of great interest [61]. LDLR is widely expressed in the brain and is able to attach to different ligands (in addition to lipoprotein metabolism) to mediate endothelial transcytosis and address the endocytic cargo to the lysosomes [59]. In this sense, LDLR-exosomes should be designed in a way to promote the activity of the LDLR subsets that mediate endothelial transcytosis rather than intracellular metabolism and protease activity. Human brain microvessels present an overexpression of INSR compared to peripheral tissues as well as brain parenchyma [69]. Conversely, TfR (the receptor responsible for the intracellular transport of transferrin) is highly expressed in human BCECs [70]. For this reason, receptor-mediated transcytosis for the BBB crossing of therapeutics is commonly performed by targeting TfR [61].

Likewise, labeling specific peptides on the surface of exosomes to be recognized by brain receptors is a widely used strategy to promote targeting and receptor-mediated transcytosis. The most used TfR-binding molecules are T7 peptide [71] and TfR-targeting antibodies [72,73]. The humanized INSR antibody (HIRMAb) [74] is the most used for INSR targeting. The most common LDLRs-binding molecules are apolipoprotein B (ApoB) and apolipoprotein E (ApoE) [75,76] or Angiopep-2 [70,77,78]. Likewise, under the typical inflammation conditions of neurodegenerative diseases, it has been reported that VCAM-1 and P-selectin receptors appear highly distributed on the luminal surface of BBB microvessels [79,80,81]. In this sense, VCAM-1 and P-selectin have also been described as possible ligand candidates for the brain targeting of exosomes.

### 5.3. Genetic Engineering

In the genetic engineering procedure, donor cells are genetically modified to force them to deliver ligand-bearing exosomes [82]. To this end, the coding sequence of the ligand is embedded in between the N-terminal of the developed peptide of a transmembrane protein and the signal peptide, and a two-step PCR is commonly used through the fusion of a reading cassette into a plasmid followed by transfection into the host cells. Different ligands, such as RAB, cofilin, tetraspanins, actin, annexin, and HSPs, among others, are necessary for surface functionalization [59]. Some of the advantages of this technique are the efficiency of manipulation and the wide variety of ligands that can be attached. This technology has successfully been utilized for exosome surface functionalization in phages and liposomes [83]. However, this procedure also presents numerous disadvantages, such as its expensive cost, the fact that many engineered exosomes cannot be distinguished in the biofluids, and the possibility of errors in the expression of ligands with high molecular weights.

### 5.4. Cell Penetrating Peptides

Cell-penetrating peptides (CPPs) are a group of short peptides that have the ability to cross cell membrane bilayers by inducing the translocation of macromolecules through unspecific interactions with the cell membranes [84,85]. They can be divided into three categories: cationic, amphipathic, and hydrophobic [84]. Cationic peptides are composed of arginine and lysine residues with positive charges that interact with negatively charged membranes. They are mainly represented by the transactivator of transcription (TAT), which has been widely investigated as an inducer of the intracellular delivery of therapeutics [86]. Amphipathic CPPs are composed of polar and nonpolar amino acid regions and are commonly found in nature [85]. Finally, hydrophobic CPPs contain nonpolar hydrophobic residues that interact with the hydrophobic domains of membranes to promote the penetration of functionalized carriers.

Cerebral capillaries are densely covered with glycocalyx, a negatively charged glycoprotein, which renders positively charged CPPs a promising surface functionalization strategy for exosomes to cross the BBB [87]. However, CPPs have widespread biodistribution in organs owing to their lack of tissue specificity and cytotoxicity properties [88], both of which are issues that must be addressed when using CPPs for brain delivery.

### 5.5. Viral Ligands

Viral proteins have aroused much interest in brain-targeting strategies since neurotropic virus-derived peptides, such as RVG, have exhibited the ability to enter into the CSF and brain parenchyma [89]. Pre-clinical studies have shown promising results by inducing brain targeting by binding RVG29 in the Lamp2b residues of the exosomes’ surface [89]. Similarly, another study showed that the systemic administration of RVG-modified exosomes in a mice model of AD led to an increased exosome amount in the brain higher than nude exosomes [90]. This accumulation resulted in the suppression of pro-inflammatory cytokines IL-6, TNF-α, and IL-β and a significant clearance of Aβ plaques [90]. This innovative approach possesses several advantages, such as the lack of bulk manipulation and formulation and lower toxicity [91]. However, the exact BBB crossing pathway of this strategy remains unknown.

### 5.6. Non-Covalent Interactions

The non-covalent strategy is mainly represented by hydrophobic, electrostatic, and protein–protein interactions. These techniques have been widely used in the modification of different nanoparticles [92], and can be transferred to the functionalization of the exosomes’ surface. The non-covalent interactions use a mix of different techniques described above to perform the linkage.

In the protein–protein interactions, CP05 peptide is commonly used to bind the CD63 protein of the exosomes’ surface. This conjugated is attached in turn to an RVG peptide, which has been shown to improve the delivery rate to the brain parenchyma, as described above [89]. Electrostatic interactions have also been employed for the targeting of the positively charged moieties of exosomes to negatively charged biological membranes. Notably, lipofectamines and cationic pullulans are the positively charged moieties most used for electrostatic interactions [93]. Interestingly, this interaction is the basis of the TAT CPP explained above. Finally, the hydrophobic–hydrophobic interactions are another sophisticated strategy used for the surface modification of exosomes. In this case, the exosome membrane is fused with specific ligand-modified liposomes, following the freeze–thaw procedure [94]. In this case, the liposome acts as a targeting vector for the exosome vesicle.

### 5.7. Hybrid Nanoparticles

Related to the previous strategy, the hybridization of exosomes with different nanoparticles is an interesting strategy for brain targeting. This cutting-edge technology has permitted, for example, the conjugation of synthetic exosomes expressing GLUT-4 with natural exosomes expressing vesicular stomatitis virus G protein. This generates a pH-responsive construct that can be used for the targeting and delivery of the hybrid composite in acid-/base-specific tissue conditions [95]. Hybrid nanoparticles of RVG-modified exosomes and gold NPs, which showed theranostic susceptibility, have also been developed for brain-disorder applications [96]. Likewise, gene–chem nanocomplexes are a novel hybridization technology that involves the modification of polymer particles with liposomes with various types of targeting molecules for brain drug delivery [97]. Although this strategy has shown some immunological issues, the hybridization of exosomes with these synthetic complexes was successfully applied to drug delivery to the brain. An interesting study carried out by Liu et al. developed an innovative gen–chem/exosome nano-scavenger that co-incorporated both hydrophobic small-molecule drugs and hydrophilic genes for the treatment of high ROS environments in PD [98].

## 6. Drug Loading of Exosomes

Depending on the chemical nature of the active substance to be incorporated in/on the exosomes, different loading methods can be used [99]. It is noteworthy that the exosome structure can be easily damaged by the loading process, so the key parameters (e.g., the power of physical interaction, the operation time, or the concentration of reagents) must be thoroughly optimized for an efficient loading [100]. According to the fabrication methods, the drug-loading of exosomes can be divided into three main categories: physical, chemical, and biological techniques (Figure 3).

Chemical methods are usually more effective and softer than other techniques. One of the main methods used is the saponin-assisted method, which requires incubation with different surfactants, such as triton or saponin, to promote drug loading by increasing the permeability of exosomes’ membranes [101]. Saponin is the most used surfactant due to its natural origin. It interacts with the cholesterol located on the exosome’s surface, which generates pores that facilitate the penetration of different substances into the exosome [102]. However, the in vivo applications of these loaded exosomes are limited mainly due to saponin’s well-known hemolytic properties [103].

Another technique for the incorporation of substances into exosomes by chemical processes is the use of transfection reagents, such as polyethylenimine (PEI), diethylaminoethyl-Dextran, liposomes, or calcium phosphate [100]. The main mechanism is based on the formation of the co-precipitates of calcium phosphate with other reagents (e.g., liposomes, nucleic acids) through electrostatic interactions, thus leading to the complexation of negatively charged nucleic acids [104,105]. For this reason, this technique is generally used for the incorporation of DNA, RNA, miRNA, or non-coding RNA. These complexes are then incubated with the cell cultures to initiate the transfection phase. This mechanism leads to the expression of the proteins coded in the transferred nucleic acids and packaged into the exosomes via the classical biogenesis pathways. A commercial kit for the direct loading of miRNAs into exosomes has been recently developed [106,107], but the residuals derived from the isolation process can affect exosome reconstitution and functionality.

Physical methods include electroporation, sonication, extrusion, and freeze–thaw, among others. In the electroporation process, the application of an external electric pulse promotes the formation of recoverable pores in the exosome membrane [108]. At this moment, the charges are incorporated and mixed with the exosomes directly (instead of the parent cells as described above), followed by a voltage application in a chilled electroporation cuvette. This voltage can range from 0.1 to 1000 kV depending on the concentration and origin of the exosomes [109]. Due to its simplicity, this technique has become one of the most frequently used methods for loading molecules into exosomes. Nevertheless, the aggregation of loaded molecules, such as DNA, siRNA, or proteins, is one of the main limitations of electroporation loading [100].

Molecule loading into exosomes using sonication is closely related to the fabrication of liposomes. Similar to the electroporation process, probe sonication causes the formation of temporary pores (or even the breakdown and reformation of naïve exosomes), which leads to the encapsulation of molecules into the vesicles by simple diffusion [110]. However, the enormous local energy applied in this process causes a significant increase in temperature, which can affect some agents and must be controlled during all of the steps.

Similar to sonication, the extrusion method is inspired by the liposome-based drug-delivery technique. This method uses polycarbonate membranes with 100–400 nm pores through which exosomes and active substances are pushed repeatedly at a controlled temperature. This process leads to a diffusion of the substances into the exosomes [111]. Extrusion has shown a good uniform size distribution with a high packing efficiency [112]. However, excessive shear stress and an intensive extrusion force may alter the properties of exosome membranes, such as the variation in the surface charge and the structure of transmembrane proteins.

The freeze–thaw method involves the repeated fusion of the lipid bilayer. The exosomes are rapidly frozen at below −80 °C, followed by thawing at room temperature. These steps are repeated for at least three cycles, but 5 to 10 cycles have demonstrated the best results [113]. The mean diameter of the obtained exosomes could vary by only 7 nm after the loading procedure. The freeze–thaw method is a relatively soft process to load proteins and miRNA. The incorporation of a sonication step is also used to improve the loading efficiency. However, this technique does not show the best encapsulation efficiency ratios [100].

Finally, of the biological methods, viral transduction-based strategies and incubation stand out [100]. In viral transduction-based strategies, adenovirus and lentivirus are commonly used as transfection vectors. Donor cells overexpress the specific genes carried by transfected viruses. This leads to the codification of proteins and loading into/on exosomes, which will be released during the secretion process [114]. A wide variety of cells are used in viral transduction, thus representing an alternative to chemical methods, which are inefficient for some cell types. Due to their stable and well-defined transfection abilities, this method is commonly used for the therapeutic applications of genetic drugs [115]. However, this technique presents some safety risks and disadvantages, such as the transmission of pathogenicity of the viruses through the exosomes, laborious and time-consuming steps, and a mechanism of transduction that is still not fully understood [116].

Regarding the incubation method, a concentration gradient is used to incorporate small molecules by passive transport through the membranes, which will finally be driven into the exosomes. This process is commonly followed by the secretion of cargo-loaded exosomes [100]. Incubation is the most direct and easiest method to load exosomes by mixing the molecules of interest with donor cells. Moreover, it can maintain exosome integrity and the activity of cargoes better than sonication [112]. The concentration and ratios of cell/exosomes/drug significantly condition the encapsulation efficiency. For example, the loading capacity of siRNA can range from 73 to 30,000 units per vesicle. Likewise, time, temperature, and volume can also affect encapsulation efficiency [117]. High temperatures increase the fluidity of the lipid membranes, thus improving the molecule loading but also promoting protein denaturation. However, this technique presents generally low encapsulation efficiency.

## 7. Exosomes as Nanomedicine-Based Therapy for Neurodegenerative Diseases

There are different nanocarriers with both natural and synthetic origins that have been developed for the treatment of a wide variety of diseases. These nanocarriers present sizes from nm to µm, different matrix compositions, loaded drugs, or biochemical properties, highlighting liposomes, lipids, polymeric and metal nanoparticles, carbon nanotubes, or dendrimers [16,118]. Exosomes represent a promising nanomedicine strategy mainly due to their ability to cross biological barriers and migrate to organs without blood supply. The molecular-specific characteristics and carrying properties of exosomes have positioned them as outstanding candidates for therapeutic purposes. In fact, exosomes by themselves or as vehicles for the delivery of drug payload(s) are being actively explored as therapeutic agents. Some of the exosomes’ advantages regarding therapeutic purposes are their biocompatibility, stability, low toxicity, and their avoidance of the immune system, allowing them to cross blood vessels (including the BBB) [119]. Moreover, exosomes have cell tropism, allowing drug delivery specificity, and are suitable for the transport of biological drugs such as proteins or nucleic acids (as short-interference RNA (siRNA) or micro-RNA (miRNA)) [89,120]. As previously described, in some cases, exosomes are modified on their surface with specific ligands, which may also enable the development of receptor-mediated tissue targeting [21]. In neurodegenerative diseases, many studies have explored the therapeutic potential of exosomes, such as in Alzheimer’s disease (AD), epilepsy, Parkinson’s disease (PD), multiple sclerosis (MS), or amyotrophic lateral sclerosis (ALS) (Table 3).

AD is the most common form of dementia, constituting up to 50–80% of cases and affecting 50 million people worldwide [137]. It is commonly diagnosed by the occurrence of significant global cognitive decline, memory loss, and the overt impairment of daily life activities. The main hypothesis for the neurotoxicity and synaptic dysfunction in AD are extracellular amyloid-β (Aβ) senile plaques and intracellular neurofibrillary tangles (NFTs) of phosphorylated tau (p-tau), although many other mechanisms involved in AD pathogenesis have been described, such as neuroinflammation, oxidative stress, or metabolic dysfunction [16].

Several authors have explored the therapeutic potential of exosomes in AD [138]. Chen et al. recently evaluated the therapeutic performance of mesenchymal stem cell (MSC)-derived exosomes in both a human neural cell culture model and an in vivo mouse model of familial AD [121]. The authors found that MSC-exosomes restored the expression of genes related to synaptic plasticity and reduced Aβ expression. In addition, their results showed that the treated mice exhibited a significant improvement in cognitive function, neuron and astrocyte impairment, and brain glucose metabolism [121]. Similarly, Cui et al. evaluated the therapeutic potential of exosomes isolated from hypoxia-preconditioned MSCs in cortical and hippocampal neuronal cultures and in transgenic APP/PS1 mice [123]. The intravenous injection of exosomes from normoxic MSCs could rescue cognition and memory and reduce plaque deposition and Aβ levels in the brain, and reduce the activation of signal transducers, the activator of transcription 3 (STAT3), and NF-κB. Exosomes from hypoxia-preconditioned MSCs significantly improved mice learning and memory capabilities and reduced plaque deposition and soluble Aβ, GFAP, Iba 1, TNF-α and IL-1β levels, as well as the activation of STAT3 and NF-κB compared to the exosomes from normoxic MSCs [123]. The same authors went a step further and surface-modified MSC-derived exosomes with RVG to evaluate their potential in the APP/PS1 transgenic mice model of AD [90]. Their obtained results demonstrated that MSC-RVG-exosomes exhibited improved targeting to the cortex and hippocampus regions. Compared with the previous study, the mice treated with MSC-RVG-exosomes showed a significant reduction in plaque deposition and soluble Aβ levels, as well as the activation of astrocytes. Likewise, MSC-RVG-exosomes improved cognitive function and reduced the expression of pro-inflammatory cytokines more than unmodified exosomes [90].

Yuyama et al. evaluated the therapeutic effects of the intracerebral administration of neuroblastoma-derived exosomes in an AD mice model [124]. The obtained results exhibited that continuous administration of neuroblastoma-derived exosomes significantly reduced soluble Aβ levels, amyloid depositions, and Aβ-mediated synaptotoxicity [124]. Related to drug loading, an interesting study performed by Qi et al. incorporated quercetin into plasma exosomes to improve the drug’s bioavailability, enhance the drug brain targeting and evaluate their therapeutic potential in an okadaic acid-induced mice model of AD [122]. For this aim, exosomes were isolated from rat’s blood using ultracentrifugation, and quercetin was loaded by several cycles of incubation and sonication in an ice-water bath. Their results showed that the loaded exosomes improved brain targeting and the bioavailability of quercetin. Likewise, the loaded exosomes significantly reduced the tau hyperphosphorylation and formation of insoluble NFTs compared to free quercetin.

Epilepsy is a neurological disease mainly characterized by an imbalance in the electrical activity of neurons, which causes recurrent and unpredictable seizures [139]. It affects 50 million people worldwide [140], which makes it the second-most prevalent neurological disease, and 30% of all patients do not respond to the available treatments. The main hypotheses of the molecular pathways involved in epileptic seizures are related to the massive influx of Ca^2+^ into neurons and the exacerbation of glutamate excitotoxicity [141]. Long et al. evaluated the therapeutic potential of intranasal-administered MSC-derived EVs in a lipopolysaccharide-induced mice model of epilepsy [125]. The authors found that EVs prevented the rise of multiple pro-inflammatory cytokines and increased the concentration of some anti-inflammatory cytokines and growth factors in the hippocampus. Interestingly, the administration of EVs after the status epilepticus greatly reduced the activation of microglia, reduced the overall loss of neurons in the hippocampus, and averted cognitive and memory impairments in the chronic phase. Likewise, this administration also promoted normal hippocampal neurogenesis and reduced hippocampal inflammation in the chronic phase [125]. Similarly, Hao et al. evaluated the neuroprotective effects of EVs from human adipose-derived MSCs (AMSCs) in an in vitro model of cortical neurons with glutamate excitotoxicity [126]. The authors found that the AMSC-treated cells showed reduced neuronal cell damage compared to the untreated control groups. Likewise, a decline was observed in the number of apoptotic nuclei and trypan-positive cells. Moreover, AMSC-derived EVs prevented a glutamate-induced decline in mitochondrial membrane potential while promoting the increase in the levels of ATP, NAD+, and NADH, as well as the ratio of NAD^+^/NADH [126].

PD is the most common motor disorder and the second-most prevalent neurodegenerative disease, affecting 1% of the population over the age of 60, or one to two individuals over the age of 60 per 1000 people at any given time [142]. It is characterized by a progressive loss of dopaminergic neurons accompanied by the degeneration of dopaminergic terminals in the striatum, thus leading to movement coordination impairments and depression, anxiety, and cognitive decline [143]. Several studies have evaluated the therapeutic potential of exosomes in PD. miRNAs from blood exosomes have been highlighted as potential targets for diagnosis and PD treatment. In that sense, Jiang et al. evaluated the effect that exosomal miRNA-137 isolated from the serum could have in a preclinical model of PD [128]. The obtained results showed that the inhibition of miRNA-137 or the up-regulation of OXR1 ameliorated PD-induced oxidative stress injury. Moreover, the inhibition of exosomal miRNA-137 with miR-137 antagomir also ameliorated PD-induced oxidative stress injury in an in vitro model. Thus, they suggested that the down-regulation of exosomal miR-137 alleviates oxidative stress injury in PD by upregulating OXR1 [128]. Li et al. explored the therapeutic potential of exosomes by suppressing the autophagy processes in the dopaminergic neurons of the *substantia nigra* [144]. In this case, exosomes serve as genetic vectors for miRNA-188-3p, which is found to be an autophagy and pyroptosis suppressor by targeting CDK5 and NLRP3 [145,146,147]. The study assessed the levels of autophagy, injury, and inflammasomes in PD mouse and cell models and found that these processes were suppressed in both models after treating them with miRNA-188-3p-enriched exosomes [144]. Yang et al. also investigated the use of exosomes as carriers of antisense oligonucleotides (ASO) to reduce α-synuclein expression in PD [127]. The knockdown ASO-based gene strategy is a reliable and well-established method for treating neurodegenerative diseases, such as PD [148]. The authors found that ASO loaded into exosomes showed high cellular uptake and low toxicity in primary neuronal cultures. Furthermore, in an α-synuclein A53T transgenic mouse PD model, the intracerebroventricular injection of exo-ASO significantly decreased the expression of α-synuclein and attenuated its aggregation, ameliorating the degeneration of dopaminergic neurons and improving the locomotor functions of the treated mice [127]. Liu et al. performed an innovative study in which they developed an exosome coating gene–chem nanocomplex for the evaluation of its therapeutic potential in both SH-SY5Y cells and MPTP-induced mouse model of PD [98]. This nanocomplex was composed of an engineered core–shell hybrid system RVG peptide-modified exosome (EXO) curcumin/phenylboronic acid-poly(2-(dimethylamino)ethyl acrylate) nanoparticle/small interfering RNA targeting SNCA (REXO-C/ANP/S). The authors demonstrated that this nanocomplex acted as a nanoscavenger for clearing α-synuclein aggregates and reducing their cytotoxicity in PD neurons. The motor behavior of the PD mice was significantly improved after REXO-C/ANP/S treatment, as well as the activation of the immune system for α-synuclein clearance due to its natural immature dendritic cell EXO coating.

MS is an autoimmune disease mainly characterized by a demyelinating process but also oligodendropathy, axonal damage, neuroinflammation, and, finally, neuronal degeneration [149]. It is considered the most prevalent chronic inflammatory disease of the CNS, affecting more than 2 million people worldwide [150]. Although MS is not categorized as a purely neurodegenerative disease, its pathological processes lead to the irreversible destruction of neural tissue [151,152,153]. Several authors have studied the therapeutic potential of exosomes in MS in relation to the demyelination process by targeting the neuroinflammation cascade. In this sense, Li et al. studied the use of bone marrow mesenchymal stem cells (BMSCs)-derived exosomes in an experimental autoimmune encephalomyelitis (EAE) rat model of MS [129]. It is known that BMSC-derived exosomes play a key role in several autoimmune diseases, but the specific mechanisms are still unknown [154,155]. In this study, exosome treatment significantly reduced neural behavioral scores, inflammatory cell infiltration into the CNS, IL-12 and TGF-α levels, and the demyelination process. In contrast, the mRNA expression levels of M2 phenotype markers IL-10 and TGF-β were significantly increased. Fathollahi et al. analyzed the therapeutic potential of MSC-derived exosomes in an EAE mouse model of MS. In this case, the aim of the study was to evaluate the effect of the intranasal administration of SEV on disease activity and antigen-specific responses [130]. The obtained results exhibited that treatment with MSC exosomes was significantly more effective in alleviating clinical scores than MSC alone. Moreover, this decrease was associated with an increase in immunomodulatory responses. Related to that, Hosseini Shamili et al. also evaluated the immunomodulatory properties in reducing the MS clinical scores of MSC-derived exosomes [131]. In this case, the LJM-3064 aptamer, which has been shown to possess a specific affinity toward myelin and re-myelination properties [156], was conjugated to the exosomes and employed as both a targeting ligand and a therapeutic agent. The results demonstrated that exosomes promoted the proliferation of oligodendroglia in the in vitro assays and reduced the inflammatory response and demyelination in the in vivo studies [131]. Finally, Pusic et al. stimulated dendritic cell cultures with low-level IFNγ exosomes containing miRNAs with anti-oxidative stress and re-myelination properties, obtaining very promising results [133].

ALS is a neurodegenerative disease that affects the nerve cells of the brain and spinal cord, causing loss of muscle control. Although the etiology of ALS is still unknown, several proteins, such as SOD1, FUS, IL-6, or p-TDP-43, have been found to be related to the progression of neuronal damage [157]. Approximately 300,000 patients are affected by this disease worldwide, and this number is expected to increase to 400,000 by 2040 [158,159]. Many research groups are currently evaluating the potential of exosomes as diagnostic and therapeutic tools in ALS. Bonafede et al. evaluated the neuroprotective effect of exosomes in several in vitro models of ALS. In 2016, the authors performed an administration of exosomes from adipose-derived stromal cells (ASCs) in an in vitro model of ALS. They aimed to demonstrate that this administration could ameliorate ALS symptoms since ASCs promote neuroprotection. [135]. The obtained results showed that, in both naïve and over-expressing mutant SOD1 NSC-34 cells, the addition of ACS-derived exosomes rescued cells from oxidative stress-induced death [135]. In 2020, they studied the influence of ASC-derived exosomes on a SOD1G93A ALS mouse model [134]. The obtained results showed that ASC-derived exosomes protected lumbar motoneurons, muscle, and the neuromuscular junction, altogether resulting in improved motor performance [134]. Similarly, Morel et al. evaluated the internalization of exosomes into the astrocytes of SOD1G93A transgenic mice and their neuronal exosomal miRNA-dependent translational regulation of astroglial glutamate transporter GLT1 [136]. Their findings showed that exosomes directly internalized into astrocytes and increased astrocyte miR-124a and GLT1 protein levels. This process significantly and selectively increased protein (but not mRNA) expression levels of GLT1 in the cultured astrocytes. Intrastriatal injection of exosomes into adult mice also reduced GLT1 protein expression and glutamate uptake levels in the striatum without reducing GLT1 mRNA levels. Moreover, miR-124a was selectively reduced in the spinal cord tissue of end-stage SOD1 G93A mice [136].

## 8. Current Limitations and Future Potential of Exosomes as Drug Delivery Systems

Despite the important advancements in the exosome field, their therapeutic applications are still in a very early stage of development. A better yield of pure exosome isolation is still required to translate these vehicles to therapeutic scenarios. This is mainly due to the relatively low release of exosomes from the donor cells [160]. Further studies are needed to improve the performance of the current isolation methods [132,161,162]. Likewise, since exosomes are widely distributed in the blood, CSF, saliva, or urine, the development of a common efficient method for isolating exosomes remains highly challenging [163]. Other limitations—such as the high equipment and reagent cost, laboratory standardization, the requirement of skilled manpower, off-targeting in healthy tissues, insufficient production of clinical-grade exosomes, batch-to-batch variation, and the difficulties in scaling up to obtain large production amounts and achieve significant robustness in the process—significantly compromise the use of exosomes as nanomedicine-based strategies [132,161].

Furthermore, not all drug-loaded exosomes bind to the targeted site, and some of them are cleared by excretion or immune system actions. The presence of MHC class I and II molecules on the surface of exosomes [164] may trigger immunogenic reactions, thus resulting in rapid clearance [165,166].

The limited loading efficiencies of the therapeutic substances pose another significant limitation for exosome-based nanomedicine therapy [65]. This limitation can be due to both donor cell contents within the exosomes that limit the cargo space for the drugs of interest and the presence of phospholipids that protect the exosomes from degradation and compromise drug release [39]. A drug’s chemical nature is also important when evaluating the limiting factors of exosome-therapy efficiency. Drug hydrophobicity and lipid composition also affect their loading into exosomes [167]. Furthermore, achieving efficient cargo loading without altering membrane integrity remains highly challenging, mainly due to the tight lipid bilayers of exosomes’ membranes [168]. In addition, and most importantly, the production of clinical-grade high-quality exosomes in higher quantities is the chief obstacle for exosome-based nanomedicine. Obtaining sterile exosomes with high therapeutic payloads remains an unapproachable goal [39].

Nonetheless, by overcoming the drawbacks discussed here, the clinical application of exosomes as a nanomedicine-based strategy may become a reality in the near future. Innovative research is focusing all its efforts on engineering exosomes with cutting-edge methods in an attempt to combat the issues described above [65]. The clarification of the in vivo behavior of exosomal formulations could facilitate their appearance in the marketplace [99]. Critical studies, such as those investigating the associated costs and profit/risks, will help the industry commit to and invest in this strategy. Relatedly, the protection of intellectual property will help reduce the risk of nanoformulation studies’ costs [169]. It is worth noting that exosome-based nanotherapy, as a pharmaceutical product, should overcome some regulatory assessments from respective agencies to be clinically introduced. As exosomes are derived from cells, it is probable that some ethical issues may arise. The latest movements toward the definition of specific guidelines in this field are likely to clarify the way in which exosomes can achieve regulatory approval and clinical application [99].

## 9. Conclusions

Exosomes are EVs involved in a wide variety of biological functions that can easily cross the BBB bidirectionally, thus connecting the central and peripheral compartments. Neurodegenerative diseases constitute a group of pathologies whose etiology remains unknown in many cases, and there are no treatments that stop the progression of such diseases. The existence of the BBB is a critical impediment to the penetration of exogenous molecules, including many drugs. These combined dynamics have positioned exosomes as a suitable strategy of great interest for nanomedicine-based applications for brain diseases. Several studies have evaluated the therapeutic potential of exosomes in major neurodegenerative diseases, such as AD, epilepsy, PD, MS, and ALS. These studies have shown promising results that highlight exosomes’ role in improving these diseases’ physiopathology and symptoms.

However, the clinical therapeutic applications of exosomes remain in a very early stage of development. Various limitations, such as laboratory standardization, exosomal off-targeting in healthy tissues, difficulties centered around scaling up, batch-to-batch variations, and the insufficient production of clinical-grade exosomes, significantly compromise the use of exosomes in patients. The future of exosome-based nanotherapy is very promising, but it is full of challenges that entail solving the described disadvantages and overcoming regulatory assessments before being introduced into the clinical world.

## Figures and Tables

**Figure 1 pharmaceutics-15-00298-f001:**
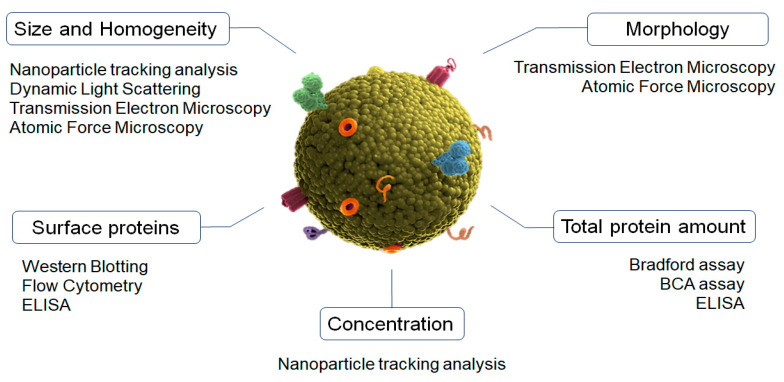
Characterization methods of exosomes main physicochemical properties.

**Figure 2 pharmaceutics-15-00298-f002:**
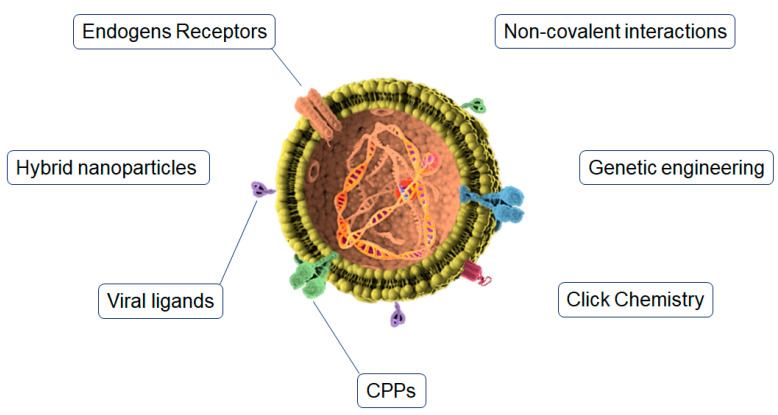
Functionalization strategies for brain targeting of exosomes.

**Figure 3 pharmaceutics-15-00298-f003:**
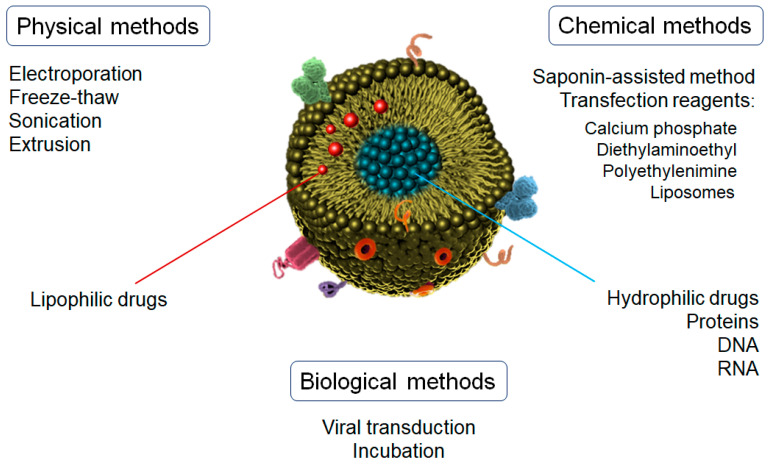
Drug-loading methods for exosomes nanoparticles.

**Table 1 pharmaceutics-15-00298-t001:** Biological-origin drug delivery systems [17,18,19].

Type of Cell/Component	Obtained from:	Favorable Properties as Drug Delivery Systems
RBC	Blood	Very large internal volume and an expandable cell surface High half-life High plasticity and strong structure Tissue infiltration capacity Simple, efficient, and highly cost-effective isolation and collection
Platelets	Blood Bone marrow	High concentration High drug loading capacity Targeting in thrombosis and hemorrhages Radiolabeling and non-radiolabeling of platelets to assess survival and recovery in injuries Efficiency of inducing cytotoxicity (great potential in cancer immunotherapy)
Stem cells	Bone marrow Skin Blood Adipose tissue Placenta	Harvested from patients, cultured and expanded in vitro, and then infused back into patients Differentiate into specialized cells (targeting) MSCs are the most used (free of ethical concerns and do not form teratomas) Low immunogenicity Immunomodulatory and re-differentiation capabilities
Macrophages	Blood	Sufficiently long blood circulation time Non-immunogenicity Phagocytic ability (considerable drug loading) Produced in high amount Easy isolation Intrinsic homing ability mediated by various cytokines (targeting)
Neutrophils	Blood	More readily available than macrophages Rapid response and high targeting at the inflammation sites Half-life of 8 h in the bloodstream Low risk of targeting and drug toxicity to normal tissues
T cells	Blood	Extensive exposure to Ag substances (enhancing the immune response and increasing immune memory time) Intrinsic biological functions with therapeutic potential (cancer cell-killing properties or recruiting of other immune cells) Great potential in cancer immunotherapy
Natural killer cells	Blood	Activation of dendritic cells and macrophages Rapidly and directly kill tumor cells without recognition Great potential in cancer immunotherapy
Adipocytes	Adipose tissue	Easy isolation and purification Good carriers for hydrophobic drugs High compatibility with the tme Good biocompatibility Long circulation Low immunogenicity
Albumin	Plasma	Average half-life of 19 days High concentration Non-toxicity Non-immunogenicity Biodegradability Stable at high temperatures, different pH and in various organic solvents Enhance the solubility of poorly water-soluble molecules Encapsulation of drugs into albumin nanoparticles or coupling of drugs to endogenous or exogenous albumin and conjugation with bioactive proteins Accumulation in malignant tissues via the EPR effect (great potential in cancer therapy) Accumulation in the arthritic articulations (great potential in rheumatoid arthritis) Cationic bovine serum albumin as a gene delivery system
Bacteria	Culture medium	Innate microbiota Biocompatible Resistance in extreme conditions (temperature, pH, O2) Alternative for the non-viral delivery Modifications to deliver drugs and genetic material to cells within tumor microenvironments

**Ag**, antigen; **RBC**, red blood cell; **MSC**, mesenchymal stem cells.

**Table 2 pharmaceutics-15-00298-t002:** Main characteristics of isolation methods of exosomes [21,22,23,24,25,26,27,28,29,30,31,32,33,34,35,36,37,38].

Technique	Isolation Parameter	Experimental Process	Advantages	Disadvantages
Ultracentrifugation	Size	A series of continuous low-medium speed centrifugation + high-speed centrifugation (100,000× *g*)	No need to label exosomes Avoid cross-contamination Low cost	High volume Time consumption Structural damage Aggregation into blocks
	Density	Density gradient thought a medium in combination with ultracentrifugation	Improvement of purity	Longer time than size
Ultrafiltration	Size	0.22 μm Filtration + ultracentrifugation	Higher sample throughput Easily adaptable High exosome purity No limitation of sample volume Low cost	Low exosomal protein yield High volume of sample Time consumption Structural damage Aggregation into blocks
Size exclusion chromatography	Size	Column filled with a gel matrix with a specific size of pores. Macromolecules penetrate along the gaps between the pores, while exosomes remain in the gel pores and are finally eluted by the mobile phase.	Exosomes enrichment High exosome purity High reproducibility High throughput (large-scale studies) No aggregation events Preservation of the biological activity and morphological integrity.	Low protein yields Co-isolation of low-density proteins High volume of sample
Immunoaffinity Chromatography	Surface ligands	Separation and purification through the binding affinity of immobilized antibodies to specific antigens on the exosomes’ surface.	Lower sample volume. High purity High yield High sensitivity Strong specificity. Qualitative and quantitative studies.	Time consumption Complicated storage conditions (not suitable for large-scale). Batch effect. Low reproducibility for proteomic analysis.
Polymer precipitation	Solubility	Reduced exosomes’ solubility with a polymer medium + low-speed centrifugation process.	More cost-effective than commercial kits, High purity and recovery rates than UC. Easy to handle Short time consuming Flexible with sample volume	Low purity and recovery rate. False positives. Co-precipitation of polymer and proteins.
Commercial Kits	Surface ligands and/or solubility	Combination of immunoaffinity and precipitation.	Easy handling Time-saving High yield Good integrity Low sample volume	Uneven extraction effect High cost Intermediate purity

**Table 3 pharmaceutics-15-00298-t003:** Selected relevant studies of recent findings of exosomes-nanomedicine based treatments for neurodegenerative diseases.

Pathology	Source of Exosomes	Isolation Method	Main Results	Ref.
AD	MSCs	Exo-Prep^®^ kit + SEC	MSC-exosomes restored the expression of genes related to synaptic plasticity and reduced the Aβ expression. In addition, their results showed that treated mice exhibited a significant improvement in cognitive function, neuron and astrocyte impairment and brain glucose metabolism.	[121]
AD	Plasma (rats)	Ultracentrifugation	Quercetin-loaded exosomes improved brain targeting and bioavailability of quercetin. Quercetin-loaded exosomes significantly reduced the tau hyperphosphorilation and formation of insoluble NFTs.	[122]
AD	MSCs	Ultracentrifugation	MSC-RVG-exosomes improved targeting to the cortex and hippocampus regions. Mice treated with MSC-RVG-exosomes showed significantly reduced plaque deposition and soluble Aβ levels, as well as the activation of astrocytes. Likewise, MSC-RVG exosomes improved cognitive function and reduced the expression of pro-inflammatory cytokines more than unmodified exosomes.	[90]
AD	MSCs	ExoQuick^®^ Kit	Exosomes from hypoxia-preconditioned MSCs significantly improved mice learning and memory capabilities and reduced plaque deposition and soluble Aβ, GFAP, Iba 1, TNF-α and IL-1β levels, as well as the activation of STAT3 and NF-κB compared to exosomes from normoxic MSCs.	[123]
AD	Neuro2a cells	Ultracentrifugation	Intracerebral administration of neuroblastoma-derived exosomes significantly reduced soluble Aβ levels, amyloid depositions, and Aβ-mediated synaptotoxicity.	[124]
Epilepsy	MSCs	SEC	Animals receiving MSC-derived EVs exhibited diminished loss of glutamatergic and GABAergic neurons and greatly reduced inflammation in the hippocampus. Moreover, the neuroprotective and anti-inflammatory effects of MSC-derived EVs were coupled with long-term preservation of normal hippocampal neurogenesis and cognitive and memory function.	[125]
Epilepsy	AMSCs	-	AMSCs-treated cells showed reduced neuronal cell damages, decreased the number of trypan-positive cells and caused a decline in the number of apoptotic nuclei. Protection by MSC-derived EVs was associated with an increased expression of GAP-43 and an elevated number of GAP-43-positive neurites.	[126]
PD	BMSCs	Ultracentrifugation	In Vitro: Exo-ASO4 also significantly attenuated α-syn aggregation induced by pre-formed α-syn fibrils. In Vivo: Exo-ASO4 intracerebroventricular injection into the brains of α-syn A53T mice significantly decreased the expression of α-syn and attenuated its aggregation. Furthermore, it ameliorated the degeneration of dopaminergic neurons in these mice and showed significantly improved locomotor functions.	[127]
PD	MSCs	Ultrafiltration + SEC	Exosomes acted as a nanoscavenger for clearing α-synuclein aggregates and reducing their cytotoxicity in PD neurons. The motor behavior of PD mice was significantly improved after exosome treatment.	[98]
PD	Serum (mice)	ExoQuick^®^-TC kit	The down-regulation of exosomal miR-137 alleviates oxidative stress injury in PD by up-regulating OXR1.	[128]
PD	ASCs	Ultracentrifugation and ExoQuick^®^-TC reagent	miRNA-188-3p-enriched exosome treatment suppressed autophagy and pyroptosis, whereas increased proliferation via targeting CDK5 and NLRP3 in PD mice and MN9D cells was observed.	[129]
MS	AMSCs	Exocib^®^ exosome isolation kit	Intranasal administration of MSC-SEV to EAE mice was more effective than the administration of MSC alone in reducing clinical scores and histological lesions of the CNS tissue.	[130]
MS	MSCs	ExoQuick^®^-TC kit	In Vitro: The aptamer-exosome promoted the proliferation of the OLN93 cell line. In vivo: The aptamer-exosome produced a robust suppression of inflammatory response as well as lowered demyelination lesion region in CNS, resulting in the reduced severity of the disease in a C57BL/6 mice model.	[131]
MS	BMSCs	Ultracentrifugation	Exosomes from BMSCs significantly decreased neural behavioral scores, neuroinflammation, and demyelination. In addition, exosomes increased the levels of IL-10 and TGF-β, whereas TNF-α and IL-12 levels decreased significantly.	[132]
MS	Dendritic cells	ExoQuick^®^ Kit	Nasally administered IFNγ-DC-Exos increased CNS myelination in vivo.	[133]
ALS	ASCs	PureExo^®^ Exosome isolation kit	ASC-derived exosomes targeted lesioned ALS regions, protected muscle, lumbar motoneurons and the neuromuscular junction, improved motor performance, and decreased glial cell activation.	[134]
ALS	ASCs	PureExo^®^ Exosome isolation kit	Exosomes were able to protect NSC-34 cells from oxidative damage and increase cell viability.	[135]
ALS	Neuronal/astrocyte primary culture	Ultracentrifugation	Exosomes directly internalized into astrocytes and increased astrocyte miR-124a and GLT1 protein levels. This significantly increased protein expression levels of GLT1 in cultured astrocytes. Exosomes also reduced GLT1 protein expression and glutamate uptake levels in mice.	[136]

**AMSCs**, Adipose-derived mesenchymal stem cells; **ASCs**, adipose-derived stromal cells; **BMSCs**, Bone marrow mesenchymal stem cells; **MSCs**, Mesenchymal stem cells.

## Data Availability

Not applicable.

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
