# Peer review of "Exosomes-Based Nanomedicine for Neurodegenerative Diseases: Current Insights and Future Challenges"

_pharmaceutics, 2023, doi:10.3390/pharmaceutics15010298_

Round 1
Reviewer 1 Report (Previous Reviewer 2)
All comments sufficiently addressed
Author Response
Thank you for your positive assessment.
Reviewer 2 Report (Previous Reviewer 1)
The revised manuscript and responses to the reviewer comments are not up to the reviewer expectation.
In the first round of review, I had asked authors to remove the information, particularly on Isolation methods, drug loading, physicochemical characterization as this information is available in plenty of published articles on the same topic. Authors didn't remove the information from these sections as I pointed out in my review report. Exploring the information already available in published reviews will not add much value to this article and may have the potential to degrade the journal standard.
Author Response
As explained in previous revision, we think that it is important to maintain these sections. We not only explored the information already available, we performed a rigorous literature search to present a critical comparison of the different techniques, which also helps the reader to contextualize and better understand the other sections.
Reviewer 3 Report (New Reviewer)
The review" Exosomes-based nanomedicine for neurodegenerative diseases: Current insights and future challenges" is in current trends.
the detailed observation and comments are as followed:
1. Format not found as per journal.
2. In Many places yellow highlights are there. Why?
3. Inclusion of clinical data, patent and market products (if any) recommended.
Author Response
- Format not found as per journal.
Journal’s format will be done when the paper is finally accepted.
- In Many places yellow highlights are there. Why?
Yellow highlights corresponds to the changes performed in the previous round of revision.
- Inclusion of clinical data, patent and market products (if any) recommended.
Thank you for this recommendation. At this moment, exosomes are in a very initial research stay and there exist several limitations for their use in humans (explained in the last section of the manuscript), so there are only pre-clinical studies in cell cultures and animal models nowadays.
Reviewer 4 Report (New Reviewer)
Dear Authors,
Congratulations to this interesting and well rounded review. I have added a file with a few comments for improvement, but otherwise find your manuscript in depth detailed and well researched.

Author Response
Thank you for your reply. We have changed/added all recommendation you did (highlighted in yellow).
Please see the attachment.

Round 2
Reviewer 2 Report (Previous Reviewer 1)
Authors have significantly improved the manuscript after the revision. Now the revised manuscript has scientifically sound information on the topic of exosomes. The added table 1 in the revised manuscript will provide a good information on biological origin drug delivery systems. The revised manuscript is now deemed to be acceptable for publication in the journal.
This manuscript is a resubmission of an earlier submission. The following is a list of the peer review reports and author responses from that submission.
Round 1
Reviewer 1 Report
Authors made an attempt to provide a review on “Exosomes-based nanomedicine for neurodegenerative diseases: from current insights to future challenges”. The manuscript is well written, organized, easy to comprehend. However, the information provided in the manuscript existed in number of previously published reviews on the topic of exosomes. Authors would need to emphasize on what information is lacking in previous reviews.
I have few points to improve the manuscript
1. In the introduction section, authors would require to compare this review with previously available literature, what is lacking in the previous reviews and what information you will be exploring this review
2. Isolation methods and physicochemical characterization of exosomes are already covered in plenty of reviews. For instance, the papers https://doi.org/10.3390/nano11061481, 10.1016/j.addr.2012.06.014 provided information on these with detailed tables, so this sections can be removed in this manuscript.
3. Again, drug loading of exosomes already exists in literature, authors can directly quote published papers for this information without explaining much on the drug loading.
4. I don’t understand the importance of Table 2. Main characteristics of drug delivery systems commonly used in nanomedicine-based treatments. There are other delivery systems, which are out of the scope of this review.
Reviewer 2 Report
Regarding the manuscript (pharmaceutics- 2103687) entitled:
“Exosomes-based nanomedicine for neurodegenerative diseases: from current insights to future challenges”
Comments to the Author
General comment
The manuscript described explores the most innovative isolation and fabrication methods, their physicochemical characterization, drug loading, the cutting-edge functionalization strategies to target them within the brain, the latest research studies in neurodegenerative diseases, and the future challenges of exosomes as nanomedicine-based therapeutic tools. The manuscript, in general, is well written and should be published after considering the following comment:
- After introduction, please add section related to biological based drug delivery like using RBCs, platelet ….etc.
1. What is the main question addressed by the research?
Using Exosomes as delivery system for neurodegenerative diseases
2. Do you consider the topic original or relevant in the field? Does it
address a specific gap in the field?
Yes, the clinical therapeutic applications of exosomes remain in a very early stage of development which require a reviewing the published studies in a regular bases
3. What does it add to the subject area compared with other published
material?
A comprehensive collection of published work
4. What specific improvements should the authors consider regarding the
methodology? What further controls should be considered?
Just addition of a new section.
5. Are the conclusions consistent with the evidence and arguments presented
and do they address the main question posed?
Yes
6. Are the references appropriate?
Yes
7. Please include any additional comments on the tables and figures.
No modification required
Reviewer 3 Report
This work places exosomes/EVs in a clinical context but unfortunately leaves open far more questions than it answers. Once something original is contributed then perhaps the paper could be given consideration for publication. Even as a review, it is not particularly well structured.
Major problems
This long article collects miscellaneous experimental findings with a promise of maybe later being assembled into something useful. Only at page 18 (section 7) it is admitted how use of exosomes is “still in a very early stage of development” where more study will be needed to “translate these vehicles to the therapeutic scenarios”.
In fact, the entire 1st paragraph here essentially undercuts the basis for the entire nanomedicine concept as currently understood.
The title itself has a typo. Consider: “Exosome-based nanomedicine for neurodegenerative diseases: Current insights and future challenges”
More disturbing is the inclusion of cadmium sulfide/selenide—both of which are toxic heavy metals with zero medical application. This topic is simply included without any disclaimer or special notice. Why are safety concerns ignored on this matter?
Figure 1 has lines directed to specific “structures” with no coherent relation to an amorphous green blob. This peculiar feature is seen with other odd cartoons included in this work.
Other concerns
Page 2: what is meant by “naturally controlled”?
Page 3: the phrase “predetermined rate” is ambiguous.
Page 3: “high complexity”: requires definition or at least discussion.
Table 1 includes “commercial kits” (final row) but this is insufficiently specific and could mean almost anything. Also, several grammar errors appear in this table.
Section 4.3: Why are none of the reference links hyperlinked here?
Page 10: “Chemical methods are usually softer…” what does this mean?
Table 3: Suggest using “condition” rather than “disease” (i.e., epilepsy is not strictly a disease).
BMSC table endnote is missing the word “cells”
Title is missing for Reference 1.
Reference 8 is given to support ongoing activity, but this is from 2017. So nothing since then?
Reference 37 is incorrectly listed.